# Influence of Woven-Fabric Type on the Efficiency of Fabric-Reinforced Polymer Composites

**DOI:** 10.3390/ma15093165

**Published:** 2022-04-27

**Authors:** Andrei Axinte, Dragoș Ungureanu, Nicolae Țăranu, Liliana Bejan, Dorina Nicolina Isopescu, Radu Lupășteanu, Iuliana Hudișteanu, Victoria Elena Roșca

**Affiliations:** 1Faculty of Civil Engineering and Building Services, “Gheorghe Asachi” Technical University of Iaşi, 43 Mangeron Blvd., 700050 Iaşi, Romania; andrei-octav.axinte@academic.tuiasi.ro (A.A.); nicolae.taranu@academic.tuiasi.ro (N.Ț.); isopescu@tuiasi.ro (D.N.I.); radu.lupasteanu@academic.tuiasi.ro (R.L.); iuliana.hudisteanu@academic.tuiasi.ro (I.H.); victoria-elena.rosca@academic.tuiasi.ro (V.E.R.); 2The Academy of Romanian Scientists, 54 Splaiul Independentei, Sector 5, 050094 Bucuresti, Romania; 3Faculty of Machine Manufacturing and Industrial Management, “Gheorghe Asachi” Technical University of Iaşi, 59A Mangeron Blvd., 700050 Iaşi, Romania; liliana-emilia.bejan@academic.tuiasi.ro

**Keywords:** satin fabrics, fiber-reinforced composite (FRP), composite laminate, Modified Genetic Algorithm (MGA), optimization

## Abstract

The greatest advantage of fiber-reinforced composite materials is the freedom to tailor their strength and stiffness properties, while the most significant disadvantage consists in their high costs. Therefore, the design process and especially the optimization phase becomes an important step. The geometry of the fabric of each lamina as well as their stacking sequence need to be carefully defined, starting from some basic geometric variables. The input parameters are the widths and the heights of the tows, the laminate-stacking sequence and the gaps between two successive tows or the height of the neat matrix. This paper is a follow-up to a previous work on using and improving an in-house software called *SOMGA* (Satin Optimization with a Modified Genetic Algorithm), aimed to optimize the geometrical parameters of satin-reinforced multi-layer composites. The final goal is to find out the way in which various types of woven fabrics can affect the best possible solution to the problem of designing a composite material, able to withstand a given set of in-plane loads. The efficiency of the composite structure is evaluated by its ultimate strains using a fitness function that analyses and compares the mechanical behavior of different fabric-reinforced composites. Therefore, the ultimate strains corresponding to each configuration are considered intermediate data, being analyzed comparatively until obtaining the optimal values. When the software is running, for each analysis step, a set of intermediate values is provided. However, the users do not have to store these values, because the final result of the optimization directly provides the composite configuration with maximum efficiency, whose structural response meets the initially imposed loading conditions. To illustrate how the *SOMGA* software works, six different satin-woven-fabric-reinforced composites, starting from plain weave (satin 2/1/1), then satin 3/1/1, satin 4/1/1, satin 5/1/1, satin 5/2/1 and finally satin 5/3/1, were evaluated in the *SOMGA* interface. The results were rated against each other in terms of the composite efficiency and the case characterized by minimal reinforcement undulation (thinnest laminate) were highlighted.

## 1. Introduction

Nowadays, fiber-reinforced-polymer (*FRP*) composites are widely used in almost every type of advanced engineering structure, due to their advantageous features such as: high specific strength and stiffness, ease of handling, corrosion resistance, low weight and non-magnetic properties [1,2,3,4,5,6]. From a structural engineering viewpoint, the properties of FRP composites are tailored by using fibers to mechanically augment the elasticity and ductility of the matrix [7,8,9,10,11].

A composite laminate is formed from two or more laminas, with the same or different fiber-orientation angles, stacked together as to ensure improved strength or stiffness properties [12,13,14,15]. Compared to unidirectional-fiber counterparts, the fabric reinforcements offer an improved strength and stiffness in two perpendicular directions [16,17,18,19]. By means of accurate designing, the properties of polymeric composites reinforced with textiles can be maximized in directions of primary importance, usually where the highest stresses occur.

The structural members made of textile-polymer composites can be very expensive; therefore, any strategy of cost reduction should be considered as a priority criterion. One way to achieve that consists of utilizing some sort of optimization tool, able to provide assistance in decision making, thus allowing to design structures with convenient properties. Aforesaid optimization can be accomplished by means of an evolutionary algorithm, such as the genetic algorithm (*GA*) [20,21,22,23].

In this paper, the authors focus on the development of a new *GA*-based computer software, adapted to solve the complex problem of optimizing 2D woven-fabrics-reinforced polymer composites. The final result is a program called *SOMGA* (Satin Optimization with a Modified Genetic Algorithm), its efficiency having already been proven, referring to the strength, stiffness and optimization of satin 5/2/1 fabric-reinforced composites [24].

To analyze and compare two composites with distinct characteristics, a finite-dimensional model (parametric) is utilized, where the geometry of the fabric and the properties of the constituent materials are considered to be the input variables. The analysis is performed at both macro-scale and meso-scale levels and the results consist of the in-plane strains $\varepsilon_{x},\varepsilon_{y},\gamma_{xy}$ of the material under a specific load case, indicating the utilized fraction of the material’s strength [25,26,27]. This data is fed into a rather-complex fitness function that basically offers a better score to the factor with the highest strain, forasmuch as the material does not reach its yielding limit. The optimal solution is determined by establishing the fittest individual from an overall population that evolves over time, simulating the classical process of natural selection [28]. During this process, the stresses and the strains are automatically computed and compared for each optimization step, and the final result is the one corresponding to the highest score. In this manner, the user no longer has to structurally evaluate each configuration and to store a high amount of stress-strain data, since it is provided directly with the configuration that meets the loading conditions and uses the lowest quantity of materials.

This work goes a step further by researching the effect that various types of woven fabric have on the strength and stiffness of the laminated composite. Using satin 2/1/1 (known as plain wave), satin 3/1/1, satin 4/1/1, satin 5/1/1, satin 5/2/1, and satin 5/3/1 as reinforcements for a fabric-reinforced E-glass epoxy multi-layered composite, the authors quantify the influence of the fabric type on the efficiency of the material.

## 2. The Repetitive Unit Cell and Its Mechanical Analysis

The analysis targets the repetitive unit cell (*RUC*), through a methodology that was detailed in a previously published article [29]. In addition, the aforementioned work provides inside information regarding the manner in which the SOMGA software performs the stress–strain analysis and processes the data related to the configuration of the RUC and the overall structural response of the composite element.

In simple terms, the optimization process uses a large number of variables and, thus, the complexity of the calculus call for a consequential computing power. To address this shortcoming, the SOMGA software (developed by the authors in Iași, Romania) involves methods from classical lamination theory (*CLT*) and finite element method analysis (*FEM*), which are used together to rapidly evaluate the mechanical characteristics of the multi-layer structure of a fabric-reinforced composite [30].

The *RUC* is defined as the smallest essential piece of a composite that keeps repeating itself over the whole material and it defines (at a meso-scale) the structure of the composite [31,32,33]. For example, the five harness satin (5/2/1) is a planar, orthogonal, square fabric made of two perpendicular tows, each consisting of a bunch of fibers. The tows laid down in the x direction of the lamina are called fills and the tows aligned at a right angle, in the y direction, are called warps; in Figure 1:
$n_{g}$—number of sub-cells in one direction of the *RUC* (harness); in the case of 5/2/1 satin, $n_{g}$ = 5;$n_{s}$—number of sub-cells between consecutive interlacing regions (shift); in the case of 5/2/1 satin, $n_{s}$ = 2;$n_{i}$—number of sub-cells in the interlacing region; in the case of satin 5/2/1, $n_{i}$ = 1.

The following geometric parameters are introduced to facilitate the modelling and the implementation of the specific geometry in the software, Figure 2:
$a_{f}$ and $a_{w}$—the tow width for fill and warp, respectively;$h_{f}$ and $h_{w}$—the tow height for fill and warp, respectively;$g_{f}$ and $g_{w}$—the gap between two consecutive tows for fill or warp, respectively;$h_{m}$ is the neat matrix thickness.


Based on the geometry specific to the unit cell, a full mechanical analysis can be performed. Starting from the predefined constituent’s characteristics (epoxy resin and E-glass fibers), the input parameters at the meso-scale level are the geometrical parameters corresponding to the *RUC*, associated with the laminate stacking sequence (*LSS*). The *RUC* characterizes the material geometry, being divided into individual primary elements, by discretization (a process specific to the *FEM*), Figure 3. The finer the gradient of discretization is, the more accurate the results are, but at the cost of increased computing power. Every primary element is then considered as an asymmetric laminate of its own, having five layers made of fill, matrix or warp and a local *LSS*, which depends on its relative position inside the *RUC*. For this substructure, the stiffness of each tow is evaluated based on its undulation, a parameter that also influences the neat matrix distribution.

Superimposing various laminas with distinct *LSS* and following the pattern, the extensional stiffness matrix increases accordingly. Knowing the elastic properties and a given load ($N_{x},N_{y},N_{xy}$), a specific strain ($\varepsilon_{x}^{0},\;\varepsilon_{y}^{0},\;\gamma_{xy}^{0}$) can be determined. This strain is altered, when necessary, by the ply angle given by the *LSS*. The stresses of constituents (warp, fill and matrix) at any given (*x*, *y*) position of each layer are then evaluated, using Equation (1).
(1)$${\left\{\begin{array}{c} \sigma_{x}\\ \sigma_{y}\\ \sigma_{xy} \end{array}\right\}}^{f,w,m}\left(x,y\right)={\left[\bar{Q}\left(x,y\right)\right]}^{f,w,m}\cdot \left\{\begin{array}{c} \varepsilon_{x}^{0}\\ \varepsilon_{y}^{0}\\ \gamma_{xy}^{0} \end{array}\right\}={\left[\bar{Q}\left(x,y\right)\right]}^{f,w,m}\cdot {\left[A\right]}^{-1}\cdot \left\{\begin{array}{c} N_{x}\\ N_{y}\\ N_{xy} \end{array}\right\}$$
where:
$\bar{Q_{i,j}}\left(x,y\right)$—the reduced stiffness matrix of the element located at the position (*x*, *y*);$A_{i,j}\left(x,y\right)$—the extensional stiffness matrix of element located at the position (*x*, *y*).


Applying the maximum-stress failure criterion for warp and fill, and the von Misses yield criterion for the neat matrix, failure analysis theory is used for every primary element [34]. Therefore, when a single element fails, the stiffness matrix of the entire composite is degraded by a second-order damage tensor and, thus, the loads are redistributed to the nearby neighboring elements [35]. More information regarding the yield criteria and degradation of the stiffness matrices can be found in the papers previously published by the authors [34,35].

Furthermore, the process is sequentially repeated until the overall strains no longer exhibit any increment, or until the composite fails. To quantify the composite efficiency for a specific set of loading, the global strains should be correlated: the higher the strain is, the more of the strength is utilized and thus, the composite is more efficient.

Although the failure criterions mentioned above are dependent only on the fiber and matrix elastic and strength properties, a general damage failure criterion is controlled not only by these properties, but also by composite manufacturing factors, such as the processing conditions and the fiber surface treatments [36,37]. Moreover, the accurate prediction of the composite true stress–strain state asks for the evaluation of some key parameters (the relative slippage displacement between the matrix interface and the debonded fibers, the true stress effect, the matrix plasticity, etc.), which define the constitutive relations of the composite outside the elastic range [38]. This approach may be applied based on a unified elastic–plastic constitutive analysis model (e.g., bridging model). However, although the bridging model involves only analytical formulae (in closed form), the implementation of such a high amount of input data in the presale version of the SOMGA software will substantially increase the computational time. In the following versions of the SOMGA software, the improvement in the accuracy and the refinement of the results through the bridging model are set as objectives.

## 3. Composite Optimization Using *SOMGA* Software

The *SOMGA* software uses a search optimization approach that is based on the classical principles of genetics and natural selection. Thus, it enables a population consisting of a large number individuals to continuously evolve by the rules of selection, a case that maximizes a specific fitness that is equivalent to minimizing a specific “*kost*” function. The latter, whose role is basically to attach a score to the composites in order to be able to rank them inside a group, is a combination of the in-plane strains ($\varepsilon_{x}^{0},\varepsilon_{y}^{0},\gamma_{xy}^{0}$), altered by some key factors. Thus, the smallest “*kost*” is granted to a composite with the highest overall strain (optimum efficiency), while a failed composite receives a high “*kost*”; see Equation (2). If the laminate does not fail, the number of plies in the *LSS* should be maintained as small as possible and the largest strain should be guided towards the direction of the highest load.
(2)$$kost=\{\begin{array}{l} -\left(\left|\varepsilon_{x}^{0}\cdot N_{x}\right|+\left|\varepsilon_{y}^{0}\cdot N_{y}\right|+\left|\gamma_{xy}^{0}\cdot N_{xy}\right|\right)-\frac{10000}{nop}\;,\;if\;no\;fail\;occurs\\ \left|\varepsilon_{x}^{0}\right|\cdot f_{x}+\left|\varepsilon_{y}^{0}\right|\cdot f_{y}+\left|\gamma_{xy}^{0}\right|\cdot f_{xy}\;\;,\;if\;composite\;fails \end{array}$$
where:
$N_{x}$, $N_{y}$ and $N_{xy}$—the external loads, applied in *x*, *y* directions, respectively, and in the *xy* plane;$\varepsilon_{x}^{0},\;\varepsilon_{y}^{0},\;\gamma_{xy}^{0}$—the strains of the material in *x*, *y* directions and in the *xy* plane;$f_{x}$, $f_{y}$, $f_{xy}$—1 if the laminate fails in the corresponding direction and 0 in any other ways;*nop*—the number of plies in the *LSS*.


The geometry of the *RUC* as well as the *LSS* are the parameters to be optimized. The number of layers is considered a key parameter and the software tries to keep it at minimum, since fewer plies means smaller manufacturing costs. The ”*kost*” function is resource consuming and slow; thus, to speed up the convergence, some particularities were imposed:a multiprocessor, multicore architecture and parallel calculus are utilized to determine the ”kost” of all individuals in a generation;every determined ”kost” is stored in a database in order to avoid performing the same calculus more than once;for the best starting point in a vast solution domain, firstly it is determined the maximum number of plies that is necessary for a laminate to avoid failure.

All internal parameters of *SOMGA*, specific to the genetic algorithms branch, are carefully chosen to match the optimization problem, as detailed in previous works [24,29].

## 4. The Starting Point

The aim is to design the optimum satin-reinforced composite that can withstand the following external loads (the loads were chosen randomly to illustrate the capability of defining a 3D force vector): $N_{x}=1800\;\left[\frac{N}{\mathrm{mm}}\right]$, $N_{y}=600\;\left[\frac{N}{\mathrm{mm}}\right]$ and $N_{xy}=300\;\left[\frac{N}{\mathrm{mm}}\right]$. In order to obtain that result, the laminate was optimized with the *SOMGA,* as previously introduced. The materials utilized were E-glass fibers (as woven fabric) and epoxy resin. The chosen volume fraction for fill and warp is 0.70. The mechanical characteristics of the constituent materials are given in Table 1.

where:
E_1_ = E_2_ = E—the longitudinal, respective transverse modulus of elasticity of the composite constituents (since both materials are isotropic, the longitudinal modulus of elasticity = the transverse modulus of elasticity);$\nu_{12}$ = $\nu_{23}$ = ν—Poisson’s ratios in (1,2) and (2,3) planes, respectively (since both materials are isotropic, the Poisson’s ratios are the same in both (1,2) and (2,3) planes);$G_{12}$—the shear modulus of elasticity in the plane (1,2);$F_{t}$— the tensile strength of the corresponding constituents;$F_{s}$— the shear strength of the corresponding constituents.


The geometrical parameters that require optimization are classified in two categories: continuous and discrete. For each parameter corresponding to the continuous category, a specific interval of possible values is predefined, while every discrete parameter receives a particular set of values. The parameters related to the *RUC* to be optimized are listed in Table 2.

The other parameters related to *LSS* are shown in Table 3.

By choosing various types of woven fabric as composite reinforcements, the authors were focused on studying the influence that the undulation has over the number of plies. Using satin 2/1/1 (known as plain wave), satin 3/1/1, satin 4/1/1, satin 5/1/1, satin 5/2/1 and satin 5/3/1 textiles, the undulation counts over the same fabric area can be found. Then, by counting the number of layers and the thickness of each composite type, an estimation of how crimping affects the efficiency of the composite is made, remembering that every material is optimized for the same loads. The fabrics utilized in this paper are shown in Figure 4.

## 5. Results and Discussion

### 5.1. Optimisation Using SOMGA for Satin 2/1/1 Fabric

In the first case, the goal is to identify the best composite with the smallest thickness, optimizing all internal geometrical parameters and starting with a plain wave as the reinforcement type; see Figure 3. The *RUC*, consisting of 2 × 2 yarns, is illustrated in Figure 5.

The number of undulations is defined as the number of times the warp yarns crimp when exchanging position with fill yarns, relative to the middle plane. For example, in the case of satin 2/1/1, the undulation count for the warp is considered to be 8 for the *RUC* zone ($UC_{RUC}$) and 98 for the whole reference area. To quantify the undulation density, the current undulation count is compared to a reference one. This reference is the count of the most-crimpy woven fabric possible, namely, the plain wave. The undulations density ($\rho_{und}$) is the undulation count for the *RUC* in relation with the plain wave count for the same *RUC* area ($UC_{RUC}^{2/1/1}$). In this particular case of a plain weave, $\rho_{und}^{2/1/1}=100\%$; see Equation (3).
(3)$$\rho_{und}^{2/1/1}=\frac{UC_{RUC}^{2/1/1}}{UC_{RUC}^{2/1/1}}=\frac{8}{8}=1.0=100\%$$

The best found solution consists of a laminate with 32 layers and the *LSS* given in Table 4.

### 5.2. Optimisation Using SOMGA for Satin 3/1/1 Fabric

In the second optimization scenario, a satin 3/1/1 fabric was chosen as composite reinforcement. Figure 6 illustrates the fabric and the corresponding *RUC*. The number of undulations in this case is 12 for the *RUC*, the textile being less crimpy than in the first case.

The resulted undulation density ($\rho_{und}^{3/1/1}$) is this time only 66%, as given by the Equation (4).
(4)$$\rho_{und}^{3/1/1}=\frac{UC_{RUC}^{3/1/1}}{UC_{RUC}^{2/1/1}}=\frac{12}{18}=0.66=66\%$$

Similar to the first case, all geometrical *RUC* and *LSS* parameters were to be optimized by the program. The algorithm was run multiple times with a population of 50 individuals per generation. The solution found consists of a laminate with only 26 layers, having the geometrical parameters listed in Table 5.

### 5.3. Optimisation Using SOMGA for Satin 4/1/1 Fabric

In this scenario, a satin 4/1/1 fabric was utilized as reinforcement. The fabric and its *RUC* are shown in Figure 7. The number of undulations in this case is 16 for the *RUC* and the undulation density ($\rho_{und}^{4/1/1}$) is 50%, Equation (5).
(5)$$\rho_{und}^{4/1/1}=\frac{UC_{RUC}^{4/1/1}}{UC_{RUC}^{2/1/1}}=\frac{16}{32}=0.5=50\%$$

The solution found consists of a laminate with 22 layers, having the geometrical parameters for the *RUC* and *LSS* given in Table 6.

### 5.4. Optimisation Using SOMGA for Satin 5/1/1 Fabric

In this case, a satin 5/1/1 fabric was utilized as reinforcement. The fabric and its *RUC* are shown in Figure 8. The number of undulations for the *RUC* is 18 (smaller than before) and the undulation density (
$\rho_{und}^{5/1/1}$) is 36%, as shown in Equation (6).
(6)$$\rho_{und}^{5/1/1}=\frac{UC_{RUC}^{5/1/1}}{UC_{RUC}^{2/1/1}}=\frac{18}{50}=0.36=36\%$$

The solution obtained is a laminate with 18 layers, having the geometrical parameters for the *RUC* and *LSS* given in Table 7.

### 5.5. Optimisation Using SOMGA for Satin 5/2/1 Fabric

In this fifth case study, the chosen reinforcement was a satin 5/2/1 fabric. The fabric and its *RUC* are shown in Figure 9. The number of undulations for the *RUC* is 20 and the undulation density ($\rho_{und}^{5/2/1}$) is 40%, as per Equation (7).
(7)$$\rho_{und}^{5/2/1}=\frac{UC_{RUC}^{5/2/1}}{UC_{RUC}^{2/1/1}}=\frac{20}{50}=0.4=40\%$$

The solution found in this case is a laminate with 15 layers, having the geometrical parameters for the *RUC* and *LSS* given in Table 8.

### 5.6. Optimisation Using SOMGA for Satin 5/3/1 Fabric

In the sixth case, the program was instructed to use a satin 5/3/1 fabric. The fabric and its *RUC* are shown in Figure 10. The number of undulations for the *RUC* and the undulation density ($\rho_{und}^{5/3/1}$) are the same as in the previous case, equal to 20; therefore, the density is 40%.

The solution found consists of a laminate with 16 layers, having the geometrical parameters for the *RUC* and *LSS* given in Table 9.

### 5.7. Overall Results and Discutions

All results are put together in Table 10. The ply thickness (*PT*) consists of the combined heights of the fill, warp and neat matrix, as can be seen in Equation (8). The laminate thickness (*LT*) is the *PT* multiplied by the number of layers.
(8)$$PT=h_{f}+h_{w}+h_{m}$$

By looking at the data from Table 10, it can be seen that the *LT* is directly related to $\rho_{und}$ in particular and to the fabric type in general; see Figure 11. The plain-weave fabric (satin 2/1/1), which is the crimpiest of all the textiles (with $\rho_{und}^{2/1/1}=100\%$), requires the thickest laminate (13.026 mm), with the largest number of plies (32). At the other end, the satin 5/1/1 fabric has the smallest undulation density $(\rho_{und}^{5/1/1}=36\%$) and the smallest *LT* (5.240 mm). In this situation, the number of layers (18) is not the smallest number from all the fabrics; this aspect can be explained by looking at the thickness of the ply, which is only 0.291 mm—the smallest number of all.

In the case of the satin 5/2/1 fabric, the optimization software found as the best solution a laminate with 15 layers, which is the smallest number of layers overall. By looking at statistics, this makes sense, because the *PT* is 0.367 mm, thus having a lamina thicker in this situation than in the satin 5/1/1 case. It is obvious that the undulation of the yarns in reinforcement influences the strength of the laminated composite. The more undulated the tows are, the more chances there are that various sections of them are not aligned in the desired direction, thus decreasing their local strength. Ideally, for maximum strength, the fibers inside tows should be always straight, but this is not possible in practice. In this situation, undulations can be closed to a minimum by using satin $n_{g}$/1/1 as reinforcement, where the harness ($n_{g}$) is five or more, as long as the fabric stays in one piece (the tows do not become loose).

As far as using satin 5/1/1, satin 5/2/1 or satin 5/3/1, there is insignificant differences between them, both in undulation (given by $\rho_{und}$) and in strength (measured by *LT*). Therefore, the shift ($n_{s}$) is not important, as their price and availability are the only factors for favoring one over the others. If the smallest number of plies is considered the most important criterion of the composite-manufacturing process, the *SOMGA* software can achieve better results by locking, from the beginning, the lamina fabric geometry, therefore imposing the same lamina thickness for all fabric types. Starting from this premise, the optimization process will focus only on *LSS*, resulting in the absolute minimum number of layers. Another solution would be to increase as much as possible the actual importance factor for the number of plies inside the “*kost*” function.

## 6. Conclusions

An efficient design process of fabric-reinforced composites must always include an optimization phase, since composites are costly materials. Having this in mind, the authors developed a new software called *SOMGA*, which was utilized to find the best solution for the optimization problem. Based on the results presented in this paper, the following conclusions can be drawn:Made in the *MATLAB*^®^ programming environment, this software is in fact an altered *GA*, capable of analyzing and optimizing a satin-reinforced composite with the help of *CLT* and *FEM*.The most important part in using this software is to fine tune the parameters such that, given the huge size of the solution domain, the program is able to determine a solution close enough to the global minimum, thus avoiding being trapped by a local minimum.The capabilities of SOMGA software were illustrated through an optimization process of six different satin-woven-fabric-reinforced composites, starting from plain weave (satin 2/1/1), then satin 3/1/1, satin 4/1/1, satin 5/1/1, satin 5/2/1 and finally satin 5/3/1.The best arrangements for the *RUC* and *LSS* geometry were found, starting from a set of external loads. Although the loads were chosen randomly, the imposed values validate the advantageous feature of the SOMGA software, enabling the definition of various loading patterns.The results were rated against each other in terms of composite efficiency, which means lower material and smaller manufacturing costs. In this regard, the less undulation the reinforcement has, the thinner the laminate can be made.As *SOMGA* is an evolutionary algorithm, distinct solutions can actually be discovered for the same problem, as a function of its complexity.The best solution is to use unidirectional fibers, having no undulation at all, but that is impossible to achieve in many cases, especially when complex shapes and molds are necessary in the manufacturing process. In those cases, utilizing satin with as little undulation as possible is a good compromise, given its good drapeability. This favors the fabrics with higher harnesses, while the shift is not important and further study should be performed to see how the interlacing fits in this equation.In terms of laminate-manufacturing complexity given, especially, by the number of composite’s layers, the finest results were determined when using satin 5/2/1. The results obtained in this case represent a 15-layer composite, having a thickness of 5.503 mm, which represents a 53% decrease in manufacturing costs when compared to the 32 layers of the standard plain weave.By default, a genetic-algorithm-based software is very resource intensive. The computational time for an element depends on the capabilities of the hardware equipment (especially the processor and the installed memory—RAM). For example, the tests described in this paper run on a commercially available 8 core processor pc, having 16 GB of RAM and the computational time for a laminate is between 8 and 24 h, depending on the number of layers.In order to speed up the operations as much as possible, all the intermediate results can be stored in a database so they can be extracted from there, if necessary, instead of being computed each time.

## Figures and Tables

**Figure 1 materials-15-03165-f001:**
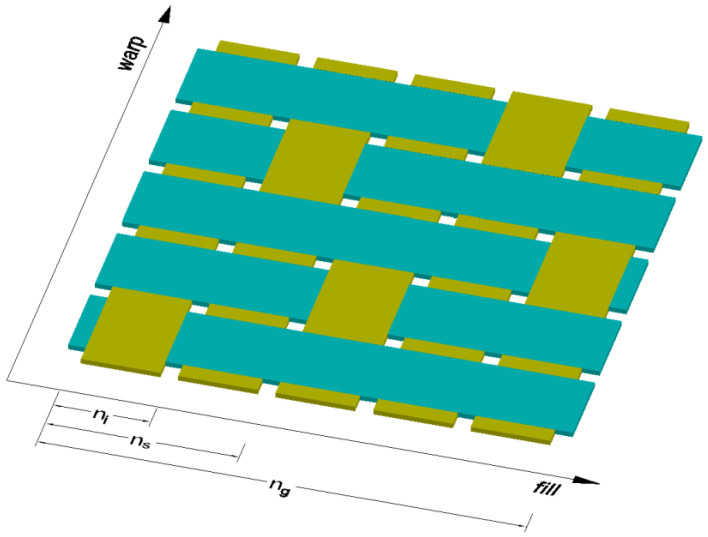
The parameters defining repetitive unit cell for satin 5/2/1 reinforced composite.

**Figure 2 materials-15-03165-f002:**
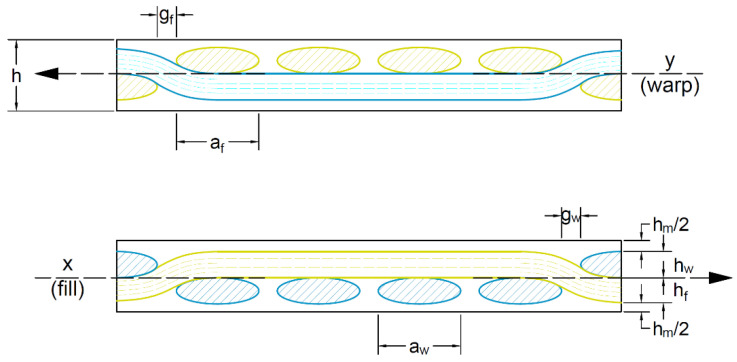
The geometrical parameters for woven-fabric-reinforced composite. The *y* and *x* axes cross-sectional views.

**Figure 3 materials-15-03165-f003:**
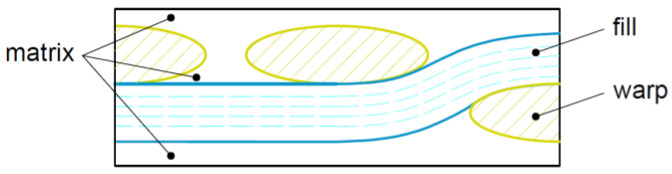
The local layers of a primary element—front view.

**Figure 4 materials-15-03165-f004:**
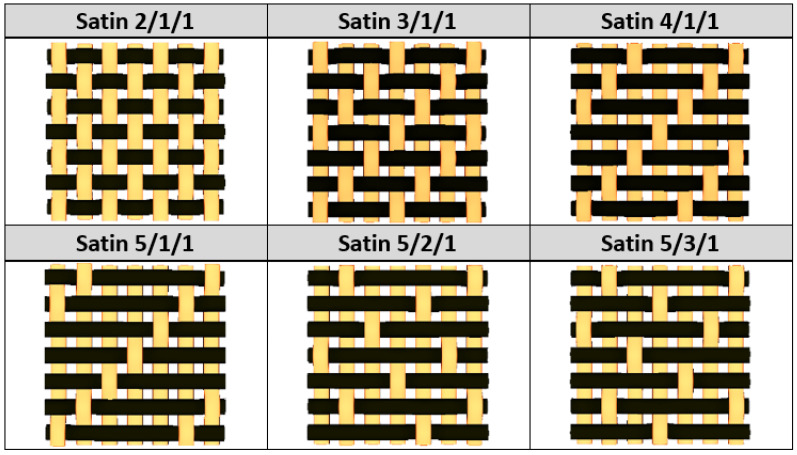
Types of fabrics utilized as composite reinforcement.

**Figure 5 materials-15-03165-f005:**
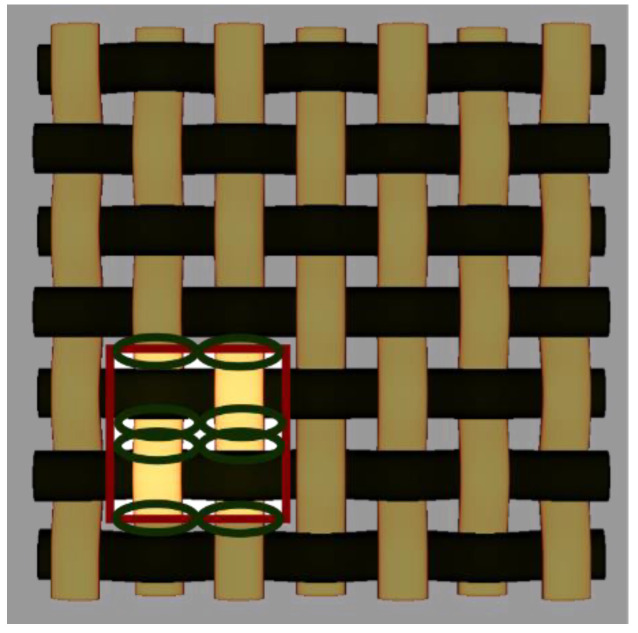
The *RUC* of satin 2/1/1 relative to a 7 × 7 yarns reference area.

**Figure 6 materials-15-03165-f006:**
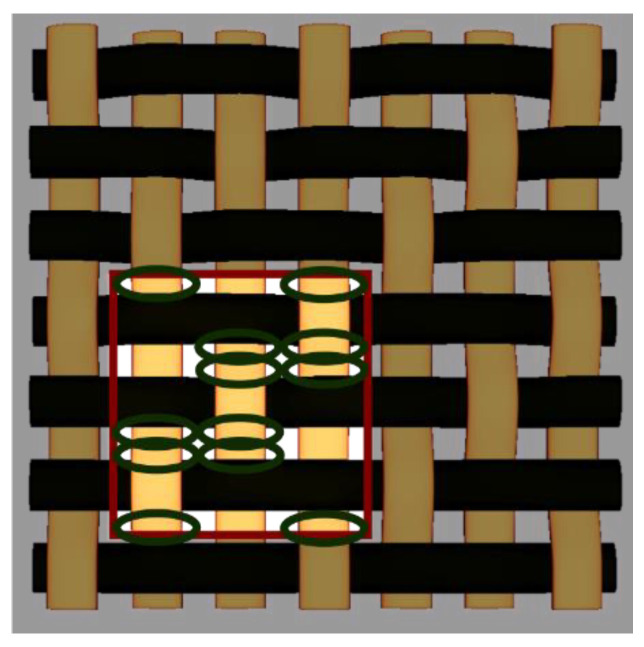
The *RUC* of satin 3/1/1 relative to a 7 × 7 yarns reference area.

**Figure 7 materials-15-03165-f007:**
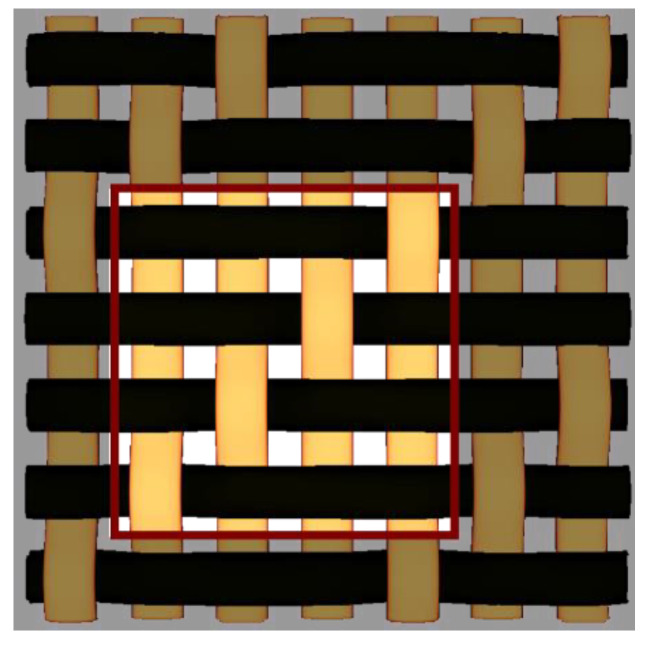
The *RUC* of satin 4/1/1 relative to a 7 × 7 yarns reference area.

**Figure 8 materials-15-03165-f008:**
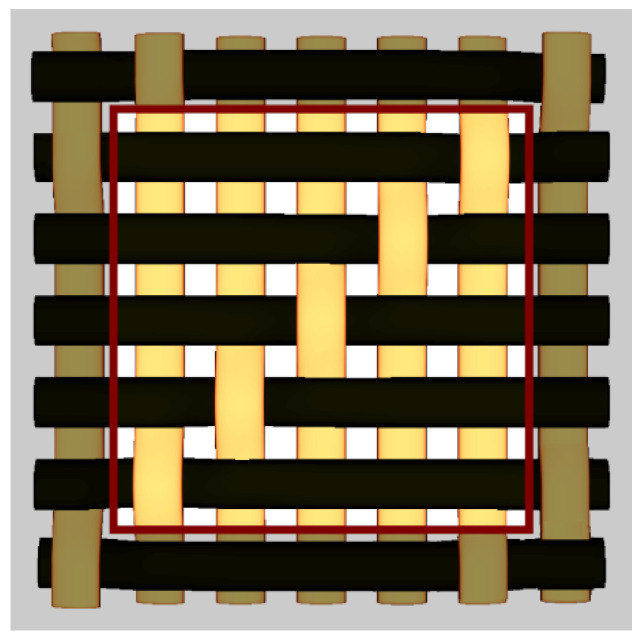
The *RUC* of satin 5/1/1 relative to a 7 × 7 yarns reference area.

**Figure 9 materials-15-03165-f009:**
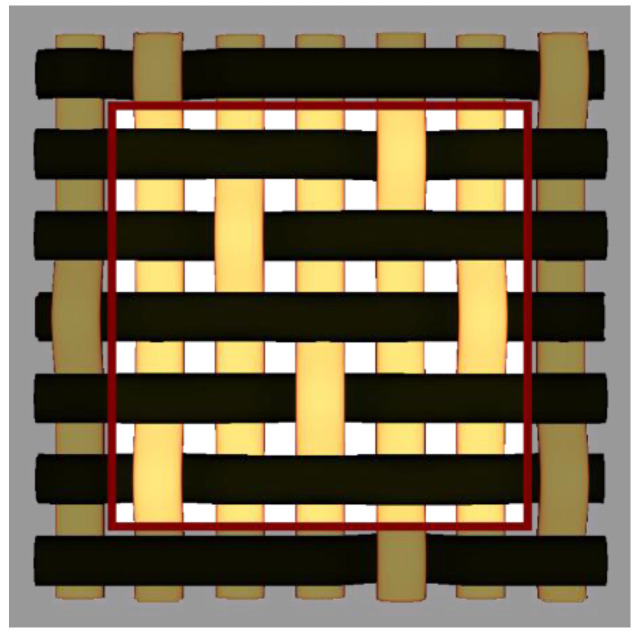
The *RUC* of satin 5/2/1 relative to a 7 × 7 yarns reference area.

**Figure 10 materials-15-03165-f010:**
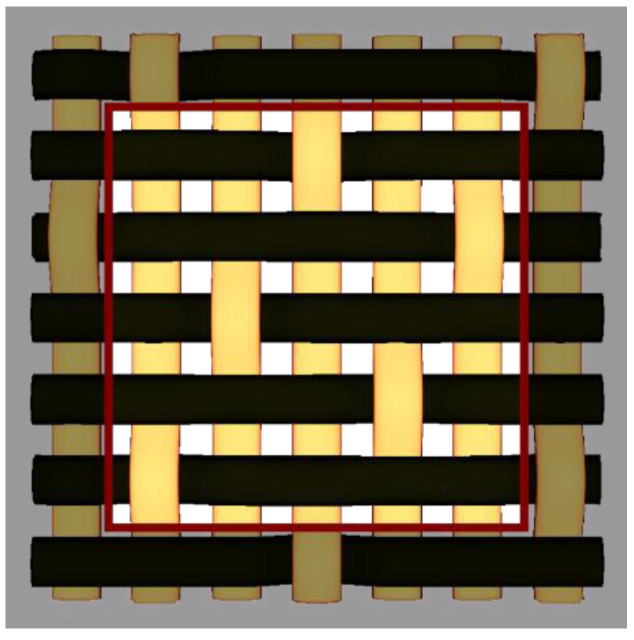
The *RUC* of satin 5/3/1 relative to a 7 × 7 yarns reference area.

**Figure 11 materials-15-03165-f011:**
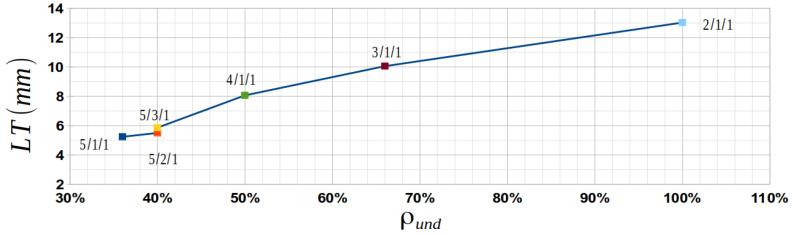
Variation in laminate thickness with undulation density.

**Table 1 materials-15-03165-t001:** The mechanical characteristics of the constituent materials [24,29].

Constituent	E_1_ = E [GPa]	E_2_ = E [GPa]	$\nu_{12}=\nu \;[\text{-}]$	$\nu_{23}=\nu \;[\text{-}]$	*G*_12_ [GPa]	*F_t_* [MPa]	*F_s_* [MPa]
E-glass fibers	72.00	72.00	0.30	0.30	27.70	1995.00	-
epoxy resin	3.50	3.50	0.35	0.35	1.30	36.60	43.00

**Table 2 materials-15-03165-t002:** Geometrical parameters related to *repetitive unit cell* to be optimized.

Parameter	Class	Type	Limit/Values	Unit
$a_{f}$	discrete	real	0.4; 0.5; 0.6; 0.7; 0.8; 0.9; 1.0	mm
$a_{w}$	discrete	real	0.4; 0.5; 0.6; 0.7; 0.8; 0.9; 1.0	mm
$h_{f}$	discrete	real	0.043; 0.063; 0.083; 0.103; 0.123; 0.143; 0.163; 0.183; 0.203	mm
$h_{w}$	discrete	real	0.043; 0.063; 0.083; 0.103; 0.123; 0.143; 0.163; 0.183; 0.203	mm
$g_{f}$	continuous	real	from 0.0 to 0.9	mm
$g_{w}$	continuous	real	from 0.0 to 0.9	mm
$h_{m}$	continuous	real	from 0.0 to 0.002	0.35

**Table 3 materials-15-03165-t003:** Geometrical parameters related to *laminate stacking sequence* to be optimized.

Parameter	Class	Type	Limit/Values	Unit
**number of plies**	discrete	integer	from 1 to 100	plies
**orientation angle**	discrete	integer	−60, −45, −30, 0, 30, 45, 60, 90	degree

**Table 4 materials-15-03165-t004:** The best solution found when using satin 2/1/1 fabric as reinforcement.

Solution for (2/1/1)	$a_{f}\;[\mathrm{mm}]$	$a_{w}\;[\mathrm{mm}]$	$h_{f}\;[\mathrm{mm}]$	$h_{w}\;[\mathrm{mm}]$	$g_{f}\;[\mathrm{mm}]$	$g_{w}\;[\mathrm{mm}]$	$h_{m}\;[\mathrm{mm}]$
*RUC*	1.000	0.800	0.203	0.203	0.144572	0.815227	0.001072
*LSS* (32 layers)	0/0/0/−30/0/0/−60/60/0/90/45/0/0/90/0/45/−30/−60/−45/−45/−45/90/0/0/60/−45/0/0/0/−30/0/30						

**Table 5 materials-15-03165-t005:** The best solution found when using 3/1/1 fabric as reinforcement.

Solution for (3/1/1)	$a_{f}\;[\mathrm{mm}]$	$a_{w}\;[\mathrm{mm}]$	$h_{f}\;[\mathrm{mm}]$	$h_{w}\;[\mathrm{mm}]$	$g_{f}\;[\mathrm{mm}]$	$g_{w}\;[\mathrm{mm}]$	$h_{m}\;[\mathrm{mm}]$
*RUC*	0.600	0.700	0.203	0.183	0.059213	0.002521	0.000781
*LSS* (26 layers)	0/−60/90/−45/−45/90/0/−30/−60/60/−45/30/−45/0/90/−30/−60/0/60/0/30/90/0/0/0/45						

**Table 6 materials-15-03165-t006:** The best solution found when using satin 4/1/1 fabric as reinforcement.

Solution for (4/1/1)	$a_{f}\;[\mathrm{mm}]$	$a_{w}\;[\mathrm{mm}]$	$h_{f}\;[\mathrm{mm}]$	$h_{w}\;[\mathrm{mm}]$	$g_{f}\;[\mathrm{mm}]$	$g_{w}\;[\mathrm{mm}]$	$h_{m}\;[\mathrm{mm}]$
*RUC*	0.600	0.900	0.183	0.183	0.000895	0.031932	0.000558
*LSS* (22 layers)	0/0/−45/0/30/30/−30/0/0/45/30/90/−60/60/90/0/60/0/−60/−30/30/−60						

**Table 7 materials-15-03165-t007:** The best solution found with satin 5/1/1 fabric as reinforcement.

Solution for (5/1/1)	$a_{f}\;[\mathrm{mm}]$	$a_{w}\;[\mathrm{mm}]$	$h_{f}\;[\mathrm{mm}]$	$h_{w}\;[\mathrm{mm}]$	$g_{f}\;[\mathrm{mm}]$	$g_{w}\;[\mathrm{mm}]$	$h_{m}\;[\mathrm{mm}]$
*RUC*	0.700	0.800	0.208	0.083	0.003727	0.027611	0.000135
*LSS* (18 layers)	0/−45/0/30/0/45/60/0/60/90/90/0/30/30/45/0/90/0						

**Table 8 materials-15-03165-t008:** The best solution found with satin 5/2/1 fabric as reinforcement.

Solution for (5/2/1)	$a_{f}\;[\mathrm{mm}]$	$a_{w}\;[\mathrm{mm}]$	$h_{f}\;[\mathrm{mm}]$	$h_{w}\;[\mathrm{mm}]$	$g_{f}\;[\mathrm{mm}]$	$g_{w}\;[\mathrm{mm}]$	$h_{m}\;[\mathrm{mm}]$
*RUC*	0.900	0.700	0.203	0.163	0.023779	0.010001	0.000854
*LSS* (15 layers)	−45/0/0/45/0/30/30/30/30/0/90/0/0/−60/0						

**Table 9 materials-15-03165-t009:** The best solution found with satin 5/3/1 fabric as reinforcement.

Solution for (5/3/1)	$a_{f}\;[\mathrm{mm}]$	$a_{w}\;[\mathrm{mm}]$	$h_{f}\;[\mathrm{mm}]$	$h_{w}\;[\mathrm{mm}]$	$g_{f}\;[\mathrm{mm}]$	$g_{w}\;[\mathrm{mm}]$	$h_{m}\;[\mathrm{mm}]$
*RUC*	0.500	1.000	0.183	0.183	0.038900	0.099765	0.000388
*LSS* (16 layers)	0/45/30/0/90/0/60/0/45/0/−45/90/30/−30/−60/0						

**Table 10 materials-15-03165-t010:** The best solution when using various fabrics as reinforcement.

Fabric Type	2/1/1	3/1/1	4/1/1	5/1/1	5/2/1	5/3/1
*ρ_und_* (%)	100(%)	66(%)	50(%)	36(%)	40(%)	40(%)
*PT* (mm)	0.407	0.387	0.337	0.291	0.367	0.366
*No. of layers*	32	26	22	18	15	16
*LT* (mm)	13.026	10.056	8.064	5.240	5.503	5.862

## Data Availability

The data underlying this article will be shared on reasonable request from the corresponding author.

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
