# Peer review of "Influence of Woven-Fabric Type on the Efficiency of Fabric-Reinforced Polymer Composites"

_materials, 2022, doi:10.3390/ma15093165_

Round 1

Reviewer 1 Report

In this manuscript, the possible usage of various wooden fabrics was investigated as a follow to a previously created GA based SOMGA software. The goal of the manuscript was to find the best solution for designing fabric reinforced polymer composites. The manuscript is well written and is of great importance in the composite field of research. Some questions and suggestions are provided below to enrich the scientific level of the presented manuscript before its publication:

  • It is suggested to add a summary of results to the last sentences of the abstract.
  • The keyword could be reviewed and sorted based on the most important level in the manuscript.
  • The literature review could be enriched by adding the following references in the Introduction section:
    • Tejyan, S., Sharma, D., Gangil, B., Patnaik, A., & Singh, T. (2021). Thermo-mechanical characterization of nonwoven fabric reinforced polymer composites. Materials Today: Proceedings, 44, 4770-4774.
    • Jahangir, H., & Esfahani, M. R. (2022). Bond Behavior Investigation Between Steel Reinforced Grout Composites and Masonry Substrate. Iranian Journal of Science and Technology, Transactions of Civil Engineering, 1-17.
    • Hasan, K. M., Horváth, P. G., & Alpár, T. (2021). Potential fabric-reinforced composites: a comprehensive review. Journal of Materials Science, 56(26), 14381-14415.
  • Figure 10 can be presented with more resolution and in a more proper format.
  • All the figures were depicted by the authors and were presented in this manuscript for the first time? Otherwise, they should be referenced.
  • Is there any figure presented from SOMGA software?
  • It is suggested to add quantitative results to the conclusion section.
  • Presenting the conclusion section in bullet format would make its reading easier.

Author Response

We appreciate your precious time in reviewing our paper and providing valuable comments. It was your valuable and insightful comments that led to possible improvements in the current version. The authors have carefully considered the comments and tried our best to address every one of them. We hope the manuscript after careful revisions meet the publication requirements of Materials Journal. The authors welcome further constructive comments if any. Below we provide a point-by-point response. All modifications in the manuscript have been highlighted in yellow.

In this manuscript, the possible usage of various wooden fabrics was investigated as a follow to a previously created GA based SOMGA software. The goal of the manuscript was to find the best solution for designing fabric reinforced polymer composites. The manuscript is well written and is of great importance in the composite field of research. Some questions and suggestions are provided below to enrich the scientific level of the presented manuscript before its publication:

 It is suggested to add a summary of results to the last sentences of the abstract.

  • As suggested by the reviewer, we have included a summary of results in the abstract section. Please see rows 33-37.

The keyword could be reviewed and sorted based on the most important level in the manuscript.

  • Thank you for pointing this out. The keywords have been reviewed and sorted based on their relevance. Please see rows 38-40.

The literature review could be enriched by adding the following references in the Introduction section:

  • Tejyan, S., Sharma, D., Gangil, B., Patnaik, A., & Singh, T. (2021). Thermo-mechanical characterization of nonwoven fabric reinforced polymer composites. Materials Today: Proceedings, 44, 4770-4774.
  • Jahangir, H., & Esfahani, M. R. (2022). Bond Behavior Investigation Between Steel Reinforced Grout Composites and Masonry Substrate. Iranian Journal of Science and Technology, Transactions of Civil Engineering, 1-17.
  • Hasan, K. M., Horváth, P. G., & Alpár, T. (2021). Potential fabric-reinforced composites: a comprehensive review. Journal of Materials Science, 56(26), 14381-14415.

  • The authors found valuable information in the articles indicated by the reviewer, which is why they enrich the Introduction section and, they cited these titles in the References section. Please see citations 9, 10 and 11.

Figure 10 can be presented with more resolution and in a more proper format.

  • Thank you for your kind reminder. We revised most of the figure captions to make them clearer. As for figure 10 (figure 11 corresponding to the revised version), minor and major axes were introduced and the format was adjusted to match the editorial requirements.

All the figures were depicted by the authors and were presented in this manuscript for the first time? Otherwise, they should be referenced.

  • We agree with the reviewer’s assessment. Accordingly, throughout the manuscript, we have revised each figure and, we have indicated the reference where needed (please see figures1-3).

Is there any figure presented from SOMGA software?

  • Thank you for pointing this out. For the development of this software, the focus was on obtaining an optimization procedure that would provide valid and reproducible results. From this point of view, the graphical interface was not considered a priority. Thus, instead of allocating time for developing a user-friendly interface, a text-based command-line was used. However, after the experimental validation of the numerical results, the development of an interface to ensure a commercial aspect is also considered.

In view of the above, at this point, the authors consider that the introduction of images depicting source lines of code is not relevant.

It is suggested to add quantitative results to the conclusion section.

  • We agree with the reviewer’s assessment. Accordingly, throughout the conclusion section, we have introduced quantitative results. Please see rows 385-389.

Presenting the conclusion section in bullet format would make its reading easier.

  • We agree with the reviewer’s assessment. Thus, the conclusion section was revised and rephrased in bullet format. Please see lines 353-398.

Reviewer 2 Report

I have some concerns regarding the novelty and contribution of your article that needs to be addressed.

 You must clearly illustrate the in-depth understanding of the relationship between structure and properties of your fabric reinforced composites.

You stated in the abstract the following sentence: The efficiency of the composite structure is evaluated by its ultimate strains using a fitness function that analyses and compares the mechanical behavior of different fabric reinforced composites. I do not find any evidence in this article about this important point. You must provide insightful information connecting your optimized six satin woven fabric reinforced composite with  its ultimate strains and must compared their mechanical properties behavior (strain, strength, and  so on.)

Without this information, your article is out of the aims and scope of this Journal. Please add this information so that your optimization process can be connected with resulting composite material properties.

Author Response

The authors appreciate the time and effort that the reviewer dedicated to providing feedback on our manuscript and are grateful for the insightful comments that provided valuable improvements to our paper. We hope the revised manuscript will suit the Materials Journal and we are available for further revisions if needed. Please see below a point-by-point response to the reviewer’ comments and concerns.

The efficiency of the composite structure is evaluated by its ultimate strains using a fitness function that analyses and compares the mechanical behavior of different fabric reinforced composites. Therefore, the ultimate strains corresponding to each configuration are considered intermediate data, being analyzed comparatively until obtaining the optimal values. When the software is running, for each analysis step, a set of intermediate values is provided. However, the users do not have to store these values, because the final result of the optimization directly provides the composite configuration with maximum efficiency, whose structural response meets the initially imposed loading conditions. Please see lines 27-37; 73-80; 88-91; 149-155

Reviewer 3 Report

Please refer to the uploaded file for all the comments.

Author Response

The submitted work deals with the optimization of the mesostructure of a composite woven fabric through a genetic algorithm. The work is believed to be of interest to the scientific community; it is well organised and clearly written. Before publication on Materials, I recommend the following points to be addressed:

The authors appreciate the time and effort that the reviewer has dedicated to providing valuable feedback on the manuscript. The comments are encouraging and the reviewer appear to share our judgement that this study and its results are of interest to the scientific community. We believe that the revised version of our paper addresses all concerns by the reviewer in detail. Please see below our detailed response to comments.

  1. The language should be improved throuout the manuscript: there are some typos (e.g.: “compared with” should be changed to “compared to”, “founded”with “found”, “stating point” with “starting point”, in the first sentence of section 2 a verb is missing…)
  • Thanks for your kind reminders. We went through the entire manuscript to eliminate grammatical mistakes. Please see the yellow highlighted lines.

  1. In section 2, an appropriate figure would be useful to show exactly the shape of the so-called “primary elements”
  • We agree with the reviewer’s assessment. Thus, a figure depicting the shape of the primary elements was added. Please see figure 3

  1. Figure 3 and Figure 7: what is called satin 5/1/1 appears to be represented as a 6/1/1. Indeed, if you take the RUC depicted in Figure 7, it is not periodic considering the represented satin. Please check
  • We agree with the reviewer’s assessment. The figures were revised. Please see figures 4 and 8.

  1. Section 2: how exactly is the stiffness of the primary elements degraded? Are multiple failure modes for the bundles considered (e.g.: transverse/longitudinal tensile/compressive or shear failure) and if so, do they lead to different stiffness degradation?
  • We agree with the reviewer’s assessment. The text was revised and more information was added. Please see lines 149-155.

  1. Section 4: how were the loads chosen? Are they taken from a specific component or just randomly? Can the GA exploit the results obtained for the present loads to run faster analyses on other loading scenario or it needs the same comuputational effort? In addition, the computational times to run the analyses should be stated.
  • Thank you for pointing this out. Although the loads were chosen randomly, the imposed values validate the advantageous feature of SOMGA software of enabling the definition of various loading patterns. By default a Genetic Algorithm based software is very resource intensive. The computational time for an element depends on the capabilities of the hardware equipment (especially the processor and the installed memory - RAM). For example, the tests described in this paper run on a commercially available 8-core processor pc, having 16GB of RAM and the computational time for a laminate is between 8 and 24 hours, depending on the number of layers. In order to speed up the operations as much as possible, all the intermediate results were stored in a database so they can be extracted from there if necessary, instead of being computed each time. Please see the revised text from lines 369-372; 390-400.

  1. Concerning the best LSS found: are symmetry and balance not given as a constraint on purpose? Their absence could lead to geometric issues due to normal-shear coupling and coupling between in plane load and bending. In addition, it could be convenient to gather some plies with the same orientation together (more convenient economically), while it seems that the GA algorithm does not take also this aspect into account. So, shall the designer make additional considerations from the output of the GA analysis?
  • We agree with the reviewer’s assessment. It is convenient to take into account the above-mentioned aspects. However, any software is developed in stages. The version presented in this paper correspond to the early stage (pre-release version). In this stage, the authors were focused on developing the algorithm in the least restrictive way and validating the correct function of the optimization process. Thus, starting from the general case, the use of a symmetrical and balanced laminate greatly simplifies the process of finding solutions and can be implemented at a later stage as a particular case.

  1. Several works in the literature show the importance of shifting and nesting phenomena on the mechanical properties of woven fabrics. Are these parameters taken into account in any way?

  • The layer shifts only have a small impact on the calculated homogenized macroscopic mechanical properties. The influence of the nesting is implicitly taken into account by the way the layers work together (by redistributing the stresses from the layers that failed to the others, considering the compatibility of the strains between adjacent layers). These information were added in the revised manuscript.

  1. The fibre volume fraction in the bundles should be mentioned. In addition, is there any reason why it was not considered as a variable in the analysis?

The fibre volume fraction was mentioned. Please see lines 200-201. Vf can be implemented as an input parameter for the optimization process, but this would increase the need for computing resources.

Reviewer 4 Report

The article generally presents an interesting solution in the area of composite material design. However, due to too many direct repetitions of content from the authors' previous articles (cited as items 15 and 16), it needs a complete overhaul. Paragraphs 2 3 4 should be rewritten in a more concise manner. The presentation part of the results obtained is interesting and well elaborated. However, the conclusion paragraph needs to be changed due to the repetition of the text contained in the authors' previous articles.

The introductory information section should definitely have been shortened and rewritten, more attention should be paid to the optimization problem.

The selection of literature items also needs to be reconsidered. At present, 63% are own citations.

Author Response

  • The authors appreciate the time and effort that the reviewer has dedicated to providing valuable feedback on the manuscript. The comments are encouraging and the reviewer appear to share our judgement that this study presents an interesting solution in the area of composite material design. We believe that the revised version of our paper addresses all concerns by the reviewer in detail.
  • The introduction section (including the indicated 2, 3 and 4 paragraph) was shortened and revised, according to the comments of all reviewers.
  • The entire manuscript was revised and new information was added. Please see the yellow highlighted text.
  • The methods and the results presented in this paper were compared with the findings of previously research works, performed both by some of the authors of this paper but, also performed by other research teams. The concerning of the reviewer regarding the Reference section was addressed in the revised manuscript. Please see the revised Reference section (new citations: 1, 3, 9, 10, 11, 13, 22, 23, 25, 26, 27, 28, 31, 32, 33).
  • By revising the manuscript and the Reference section, according to the reviewer assessment, the authors believe that the results of this study are now better highlighted and reported to the findings of other research teams.
  • The Conclusions section was entirely revised and rewritten.

Round 2

Reviewer 2 Report

The authors have address my comments and linked their results with the Journal scope. Therefore, it is suitable for publication.

Author Response

We appreciate your precious time in reviewing our paper and providing valuable comments. It was your valuable and insightful comments that led to the improvements in the current version, according to the publication requirements of Material Journal. 
